# Pericardial Involvement in Severe COVID-19 Patients

**DOI:** 10.3390/medicina58081093

**Published:** 2022-08-12

**Authors:** Mihai Lazar, Ecaterina Constanta Barbu, Cristina Emilia Chitu, Ana-Maria-Jennifer Anghel, Cristian-Mihail Niculae, Eliza-Daniela Manea, Anca-Cristina Damalan, Adela-Abigaela Bel, Raluca-Elena Patrascu, Adriana Hristea, Daniela Adriana Ion

**Affiliations:** 1Faculty of Medicine, University of Medicine and Pharmacy Carol Davila, No. 37, Dionisie Lupu Street, Sector 2, 020021 Bucharest, Romania; 2National Institute for Infectious Diseases Prof. Dr. Matei Bals, No. 1, Calistrat Grozovici Street, District 2, 021105 Bucharest, Romania

**Keywords:** pericardial effusion, pericarditis, SARS-CoV-2, risk factors, COVID-19 complications

## Abstract

*Background and Objectives*: SARS-CoV-2 has an extensive tissue tropism due to its ability to attach to the surfaces of cells through different receptors, leading to systemic complications. In this article, we aim to present the prevalence of pericardial effusions in patients with severe COVID-19, to identify the risk factors/predictors for pericardial involvement, and to evaluate its impact on overall mortality. *Materials and Methods*: We enrolled 100 patients with severe COVID-19 in our observational cohort study and divided them in two groups: Group A (27 patients with pericardial effusion) and Group B (73 patients without pericardial effusion). We recorded demographic and lifestyle parameters, anthropometric parameters, clinical parameters, inflammation markers, respiratory function parameters, complete blood count, coagulation parameters, and biochemical serum parameters. All patients were evaluated by computer tomography scans within 48 h of admission. *Results*: The median age was 61 years in both groups and the male/female ratio was 3.5 vs. 2.8 in Group A vs. Group B. We identified mild pericardial effusion (3–4 mm) in 62.9% patients and moderate pericardial effusion (5–9 mm) in 37.1% patients, with a median value of 4 [3;6] mm. The patients with pericardial effusion presented with higher percentages of obesity, type-2 diabetes mellitus, arterial hypertension, and congestive heart failure, without statistical significance. Increased values in cardiac enzymes (myoglobin, CK, CK-MB) and LDH were statistically associated with pericardial effusion. The overall mortality among the participants of the study was 24% (24 patients), 33.3% in Group A and 20.8% in Group B. *Conclusions*: Pericardial effusion has a high prevalence (27%) among patients with severe forms of COVID-19 and was associated with higher mortality. Pericardial effusion in our study was not associated with the presence of comorbidities or the extent of lung involvement. Overall mortality was 60% higher in patients with pericardial effusion.

## 1. Introduction

COVID-19 represents a systemic disease with a variable range of severity, from the asymptomatic clinical form to respiratory failure and increased mortality. SARS-CoV-2 has a broad tissue tropism due to its ability to attach to the surfaces of host cells through a variety of receptors, such as the angiotensin-converting enzyme 2 (ACE2), neuropilin-1, the tyrosine protein kinase receptor UFO (AXL), and antibody–FcγR complexes [1], leading to complications (hematologic, neurologic, digestive, and cardiac). Pericarditis represents a frequently under-diagnosed complication, associated with significantly higher all-cause mortality in patients with COVID-19 [2]. The current data on pericardial involvement in SARS-CoV-2 infection are scarce, being based mainly on case reports, case series of patients with clinical suspicion of cardiac involvement, or post mRNA-vaccination studies. In this article, we aim to present the prevalence of pericardial effusions in patients with severe COVID-19, to identify the risk factors/predictors for pericardial involvement, and to evaluate its impact on overall mortality.

## 2. Materials and Methods

In our observational cohort study, we enrolled 100 patients with severe COVID-19, admitted to our department between April 2020 and December 2020, and divided them in two groups: Group A (27 patients with pericardial effusion) and Group B (73 patients without pericardial effusion).

We considered a severe form of COVID-19 to be when the patient had at least one of the following criteria: peripheral oxygen saturation (SpO_2_) ≤ 93% in the ambient air, respiratory rate (RR) > 30/min, arterial oxygenation partial pressure to fractional inspired oxygen ratio (PaO_2_/FiO_2_ ratio) < 300, or lung infiltrates > 50% of the lung parenchyma [3].

All patients were diagnosed with COVID-19 following a positive real-time–polymerase chain reaction test (RT–PCR).

We recorded demographic and lifestyle parameters (age, sex, and smoking status), anthropometric parameters (weight, height, and body mass index (BMI)), clinical parameters (heart rate, systolic and diastolic blood pressure, and consciousness status), inflammation markers (C-reactive protein (CRP), interleukin 6 (IL-6), serum ferritin, white blood cells (WBC)), respiratory function parameters (respiratory rate, SpO_2_ by pulse oximetry and/or SpO_2_ by arterial blood gas analysis, along with the PaO_2_ and PaO_2_/FiO_2_ ratio), complete blood count, coagulation parameters (D-dimers, plasminogen activator inhibitor-1(PAI-1), prothrombin time (PT), and international normalized ratio (INR)), and biochemical serum parameters (brain natriuretic peptide (BNP), troponin I (TnI), creatine kinase including the MB isoform (CK, CKMB), myoglobin, serum albumin, lactate dehydrogenase (LDH), alanine transaminase, aspartate transaminase (ALT, AST), and creatinine). All patients were evaluated by computer tomography (CT) scans with a 64-slice definition AS (Siemens Healthcare GmbH) within 48 h of admission. The patients were examined in inspiratory breath-hold and supine positions. A dedicated diagnostic software, syngo Pulmo3D, was used for the quantitative evaluation of the lung parenchyma. The following factors were considered: alveolar lesion—the lung areas with densities higher than 0 Hounsfield units (HU), mixt lesions (alveolar and interstitial)—the lung areas with densities between 0 and −200 HU, interstitial lesions—the lung areas withdensities between −200 and −800 HU, and normal parenchyma—the lung areas with densities between −800 and −1000 HU [4,5]. We measured the thickness of the pericardial effusion in the axial, sagittal, and coronal planes, and the maximum thickness was registered. The imaging review was blinded, and the radiologist was unaware of other parameters registered for the patient.

To establish the diagnosis of pericarditis, we applied the European Society of Cardiology guidelines for the diagnosis and management of pericardial disease [6].

Pericardiocentesis was not necessary in any of the patients. Echocardiography was not available at the timeof the study. We excluded patients under 18 years, cases of pregnancy, and patientswith chronic pericarditis.The informed consent form was signed upon admission to the hospital.

For the statistical analysis, we used the Statistical Package for Social Sciences (SPSS version 25, IBM Corp., Armonk, NY, USA). Patient data are presented as medians and quartiles (Q1, Q3) for the continuous variables and as percentages for the categorical variables. We performed binary logistic regression using pericardial effusion as the dependent variable and the demographic, clinical, biologic, and imaging data as independent variables to evaluate the inter-parametric associations and to estimate the odds ratio (OR). We used the OR to calculate the risk of pericardial effusion for the independent variables. For OR > 1, the correlation between the pericardial effusion and independent variable was proportional, and the pericardial effusion rate was calculated using the formula (OR−1) × 100%. To calculate the “50% rate,” we used the formula 0.5/(OR−1). The statistical significance of the regression model was estimated by the Omnibus test of model coefficients and verified by the Hosmer–Lemeshow test. A *p*-value lower than 0.05 was considered statistically significant. We calculated the relationship between the pericardial effusion and the registered patient data using the Spearman correlation and ANOVA test for statistical significance [5]. The study was approved by the Ethics Committee of the hospital.

## 3. Results

In Group A (patients with pericardial effusion) (Figure 1), 27 patients with a median age of 61 [49;68] years were included; in group B (patients without pericardial effusion), we included 73 patients with a median age of 61 [51.5;66.5] years. The male/female ratio was 3:1 (75:25) for the entire study, with a higher ratio in Group A (3.5:1) and a lower ratio in Group B (2.84:1).

The pericardial effusion had a median value of 4 [3;6] mm. We considered a mild effusion to have occurred in 62.9% cases (3–4 mm) and a moderate effusion to have occurred in 37.1% cases (5–9 mm). Only eight patients with pericardial effusion met the criteria for pericarditis. Five patients with positive criteria for pericarditis also presented elevated TnI and newly installed tachyaritmia, or atrioventricular block; therefore, we considered the diagnosis of myopericarditis to be attributable. The patients with pericarditis had a medium pericardial effusion of 7.3 mm, similar to the patients with myopericarditis (5.2 mm). None of our patients presented with cardiac tamponade.

The patients with pericardial effusion presented higher percentages of obesity, type-2 diabetes mellitus, arterial hypertension, and congestive heart failure (Table 1), although we found no correlation with statistical significance between a specific comorbidity and the pericardial effusion.

Patients with pericardial effusion had higher respiratory rates, coagulation parameters (D-dimers, PAI-1), leukocytes, inflammatory parameters (CRP, serum ferritin), and cardiac markers (myoglobin, CK, CK-MB, BNP, and LDH). We found no significant differences in the lung involvement between the two study groups (Table 2).

We identifiedpleural effusion in 21 patients: 5 cases (18.5%) in Group A (ranging from 5 to 22 mm) and 16 cases (21.9%) in Group B (ranging from 3 to 65 mm).

The overall mortality in the study was 24% (24 patients), with a higher percentage in Group A (33.3%) compared with Group B (20.8%).

The patients faced a 1.5-times increased risk of developing pericardial effusion for every increase in the respiratory rate with 6 breaths/minute, in platelets with 166 × 10^3^/µL, in myoglobin with 62.5 (ng/mL), in CK, and in LDH with 166(U/L), respectively.

The occurrence of pericardial effusion presented the highest proportional correlation with myoglobin and CK. In addition to the evaluation by logistic regression, the Spearman correlation test demonstrated a correlation of pericardial effusions with CRP levels (Table 3).

Patients with pericardial effusion had a higher ICU admission rate (14.8% vs. 4.1%) and higher need for ventilation (Table 4).

## 4. Discussion

Pericardial involvement can present as an asymptomatic pericardial effusion or may be associated with symptoms or signs of myocarditis. Up to 80% of pericarditis cases in developed countries are idiopathic, presumably most often caused by viral agents. SARS-CoV2 pericarditis probably follows the same pathophysiological path as other viruses with cardiac tropism [7].

HIV infection can be correlated with a wide range of comorbidities, from immune disorders to endocrine and secondary infections, the impairment of the central nervous system, imbalances in bodycomposition, and disorders of the cardiovascular and renal systems [8,9] which increase the risk of pericardial effusion. Chronic immune activation and a persistent inflammatory state, with increased blood levels of the inflammatory cytokines–TNFα, IL-6, and CRP [10], associated with a direct cardiac involvement may induce pericardial effusions, with a lower occurrence among patients under antiretroviral therapy [11].

The cardiac involvement in patients with influenza infection more frequently consisted of myopericarditis (77.3%) than isolated pericarditis (22.7%). Tamponade was reported more frequently identified in patients with isolated pericarditis (41.2%) than in those with myopericarditis (13.8%), while the patients with myopericarditis more often presented with cardiogenic shock (64%) [12]. According to Spoto S. et al. [13], pericarditis without myocardial involvement has rarely been reported in patients infected with influenza virus. In our study on COVID-19 patients, we found a higher percentage of isolated pericarditis (37.5%) and a lower percent of myopericarditis (62.5%) compared to patients with influenza infection, and no cardiac tamponade.

The reportedprevalence of pericardial effusion in COVID-19 patients varies between 4.6% [14] and 90.7% in critically ill patients [15] and was reported as an incidental finding on CT scans [16].

Although SARS-CoV-2 infection is considered to be more severe in patients presenting with comorbidities such as obesity, diabetes mellitus, arterial hypertension, and cardio-vascular or pulmonary disorders [17], we did not find an association between the occurrence of pericardial effusion and these comorbidities.

Cardiovascular complications such as arrhythmias, hypoxemic cardiomyopathy, myocardial infarction, pericarditis, and myocarditis are more frequent in COVID-19 patients, especially in severe forms of the disease [18]. In our study based on severe COVID-19 patients, we found a prevalence of 27% of pericardial effusion, with only 8% of patients meeting the criteria for pericarditis and 5% meeting the criteria formyopericarditis. The small percentage of patients meeting the criteria for pericarditis might be explained by the low, either mild (62.9%) or moderate (37.1%), severity of pericardial effusion.

The main pathogenic mechanism leading to pericardial effusion in COVID-19 patients is considered to be a systemic inflammatory reaction, with secondary cardiac involvement. Elevated levels of IL1 and TNF-α in the inflammatory cascade are predisposing factors to pericardial inflammation [19]. Pericarditis and myopericarditis may also occur as a direct viral injury, facilitated by the presence of ACE2 in the heart, cardiomyocytes, and epicardial adipose tissue near the visceral pericardium [20]. The viral presence in the pericardial effusion was demonstrated by some authors, who detected SARS-CoV-2 in the pericardial liquid [21], while others did not detectviral RNA in the pericardial liquid [22]. In patients with severe forms of COVID-19, hypoxia occurs frequently and may also lead to pulmonary hypertension and secondary pericardial effusion [16].

In our study on patients with severe COVID-19 and extensive lung involvement, we found similar degrees of lung injuryand similar values of respiratory parameters in both groups (patients with and without pericardial effusion). Similarly, in a brief review, no association between the degree of lung involvement and the development of pericardial effusion was observed [7]. COVID-19 pneumonia has characteristic patterns of distribution, with higher involvements of the peripheral lung areas, adjacent to the pleura, associated with an increased inflammatory process. However, the percentage of patients who developed pleural effusions remained lower than those with pericardial effusions (21% vs. 27%), and only 18.5% cases of pleural effusion were associated with pericardial effusion.

Patients with COVID-19 are at risk of developing cardiac complications, such as pericardial effusion, myocarditis, myocardial infarction, arrhythmias or thromboembolic events. For COVID-19 patients with a cardiac injury, an increase in the biological markers has beenassociated with a poor prognosis [23]. In patients with severe forms of COVID-19, the increase in the serum myoglobin has been considered more significantthan elevated TnI, being associated with increased mortality [24]. In the present study, we also observed that an increased serum myoglobin is linked with the presence of pericardial effusion, with a higher correlation coefficient than other cardiac markers (CK and CKMB). The significant association of pericardial effusion with increased CRP levels indicates that pericardial inflammation plays an important role in the development of pericardial effusion in COVID-19 patients. Pericarditis is associated with high odds of mortality (2.55) and theonset of new cardiac complications (cardiac arrest, incident heart failure, and atrial fibrillation), which are higher than the odds in COVID-19-associated myocarditis [2]. Some studies suggest that pericarditis could be used as a marker forsevere diseaseand a poor outcome [7]. In a study on 530 patients hospitalized with different severity levels of COVID-19, it was demonstrated that pericardial effusion was a common finding, rarely due to acute pericarditis or myocarditis, but potentially associated with myocardial dysfunction and excess mortality [25]. We found that the overall mortality was 60% higher in patients with pericardial effusion (33.3% vs. 20.8%). Pericardial effusion was associated with myocarditis in 18.5% of cases. Although pericardial effusion may progress into cardiac tamponade [26], none of our patients presented such a complication.

In regard to pericarditis, the bacterial etiology should be considered in cases of immunodeficiency or existent pericardial comorbidities, as most bacterial pericarditis occurs either by proximity contamination or by hematogenous dissemination. A primary bacterial pericarditis without an extracardiac source is considered very rare, while a viral cause is considered the most common cause of pericarditis [27,28]. Bacterial vs. viral pericarditis is more likely associated with pericardial tamponade, and pericardiocentesis is necessary when purulent pericarditis is suspected. The occurrence of bacterial complications may be followed by septic shock with multiorgan failure, pleural effusions, and the re-accumulation of pericardial fluid or septic emboli [28].

In COVID-19, pericarditis might be underdiagnosed, resulting in patients not receiving the appropriate treatment [7]. The management of COVID-19 pericarditis is particularly challenging because there are equivocal or incomplete data regarding the classical treatment options, such as highdoses of non-steroidal anti-inflammatory drugs (NSAIDs) or colchicines [18]. As all patients included in our study were treated with dexamethasone for severe COVID-19 pneumonia, adding NSAIDs was not considered as an option, due to the cumulative sideeffects. It is possible that only a particular phenotype of COVID-19 pericarditis, which is yet to be determined by further studies, may benefit from colchicine treatment or NSAID–colchicine association, as illustrated in certain studies or clinical cases [7].

Our study had certain limitations. Firstly, chocardiography was not available at the timeof the study. Secondly, pericardiocentesis was not performed on any of the patients.

## 5. Conclusions

Pericardial effusion had a high prevalence (27%) in patients with severe forms of COVID-19, although, in most cases, the severity of pericardial effusion was mild. Only 30% of the pericardial effusions developed in the clinical context of pericarditis. We found that increased values in cardiac enzymes (myoglobin, CK, CK-MB), LDH, platelets, and CRP were associated with the presence of pericardial effusion, with myoglobin showing the highest correlation index. Pericardial effusion in our study was not associated with comorbidities or the extent of lung involvement. Overall mortality was higher in patients with pericardial effusion.

## Figures and Tables

**Figure 1 medicina-58-01093-f001:**
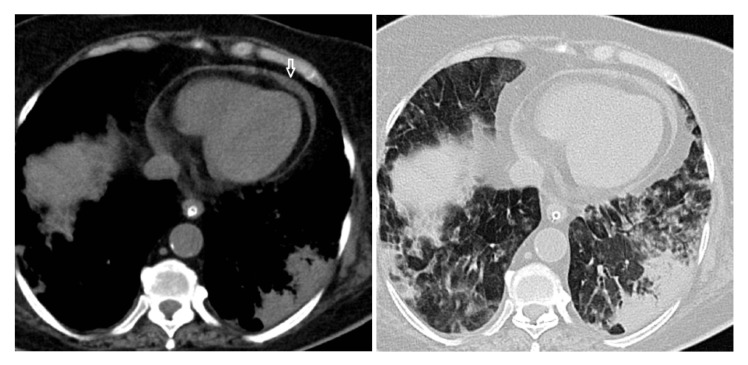
Pericardial effusion (white arrow) in a patient with a severe form of COVID-19 pneumonia.

**Table 1 medicina-58-01093-t001:** Comorbidities in SARS-CoV-2 patients with and without pericardial effusion.

Comorbidity	Group A (*n*, %)	Group B (*n*, %)	*p*-Value
Obesity	15 (55.5%)	31 (42.4%)	0.375
Diabetes mellitus type-2	7 (25.9%)	14 (19.2%)	0.334
Arterial hypertension	16 (59.2%)	29 (39.7%)	0.094
Congestive heart failure	4 (14.8%)	2 (2.7%)	0.164
Peripheral vascular disease	1 (3.7%)	1 (1.3%)	0.653
Chronic kidney disease	4 (14.8%)	4 (5.5%)	0.105
Chronic obstructive pulmonary disorder	3 (11.1%)	9 (12.3%)	0.851
Chronic viral hepatitis	1 (3.7%)	3 (4.1%)	0.907
History of neoplasia	2 (7.4%)	4 (5,5%)	0.427
History of ischemic stroke	2 (7.4%)	5 (6.8%)	0.733
Dementia	1 (3.7%)	1 (1.3%)	0.653
History of peptic ulcer	0 (0%)	1 (1.3%)	1

**Table 2 medicina-58-01093-t002:** Clinical, laboratory, and imaging characteristics in SARS-CoV-2 patients with and without pericardial effusion.

Clinical, Laboratory, and Imaging Characteristics	Group A (Median, Q1, Q3)	Group B (Median, Q1, Q3)	*p*-Value	ExpB [CI]
Consolidation (% from total lung volume)	1.7 [1.1; 4.9]	2.1 [1.1; 3.4]	0.878	
Mixt lesions (% from total lung volume)	5.1 [2.5; 9.1]	4.9 [2.9; 8.1]	0.263	
Interstitial lesions (% from total lung volume)	45.1 [36.1; 50.4]	40.3 [31.9; 52.6]	0.159	
Normal pulmonary densities (% from total lung volume)	48.4 [30.3; 56.8]	52.7 [34.3; 64]	0.165	
Total pulmonary lesions (% from total lung volume)	51.6 [43.2; 69.7]	47.3 [35.9; 65.6]	0.165	
Heart rate (beats/min)	80 [67; 89]	79 [65; 88]	0.674	
Systolic blood pressure (mmHg)	130 [108; 140]	125 [111; 137.5]	0.635	
Diastolic blood pressure (mmHg)	75 [68; 83]	75 [66.5; 82]	0.607	
Respiratory rate (breaths/minute)	27 [16; 34]	24 [20; 30]	**0.03**	1.078 [1.006; 1.156]
Saturation (O_2_)	95 [93; 97]	95 [94; 97]	0.764	
Median oxygen flow (L/min)	14 [10; 16]	14 [10; 24]	0.912	
FiO_2_ (%)	60 [50; 60]	60 [50; 60]	0.819	
Arterial O_2_ pressure (PaO_2_)	86 [69; 99]	81.5 [69.5; 104]	0.776	
PaO_2_/FiO_2_ ratio	150 [116.6; 210]	148 [110.8; 216.7]	0.692	
PCO_2_ (mmHg)	38 [34.9; 39]	36.5 [34; 40]	0.858	
pH	7.43 [7.39; 7.45]	7.45 [7.42; 7.47]	0.124	
WBC (×10^3^/µL)	8.8 [6.8; 13]	8.3 [6.5; 11.1]	0.262	
Neutrophils (×10^3^/µL)	6.5 [4.8; 11.2]	6.4 [4.8; 8.9]	0.635	
Lymphocytes (×10^2^/µL)	8 [4; 11]	8 [6; 11]	0.771	
Neu/Ly ratio	9.6 [5.3; 14.3]	7.5 [4.5; 14]	0.572	
Platelets (×10^3^/µL)	281 [211; 345]	215 [166.5; 294]	**0.05**	1.003 [1; 1.007]
Hemoglobin (g/dL)	13.5 [12.3; 14]	13.8 [12.6; 14.5]	0.202	
CRP (mg/L)	97 [40; 162]	56 [17.7; 93.4]	0.112	
Serum ferritin (ng/mL)	1331.7 [370.2; 1650]	827.8 [325.1; 1635.6]	0.377	
IL6 (pg/mL)	13.6 [3.7; 55.4]	16 [2.7; 102]	0.394	
PAI-1 (ng/mL)	500.5 [321; 708.7]	479 [284.7; 738.1]	0.993	
BNP (ng/L)	170 [16.5; 1149.5]	92.5 [22.7; 372.2]	0.143	
TnI (ng/mL)	0.03 [0.03; 0.05]	0.3 [0.3; 0.3]	0.275	
Myoglobin (ng/mL)	266 [135.5; 328.7]	108.5 [84.7; 168.9]	**0.001**	1.008 [1.003; 1.013]
D-dimers (ng/mL)	285 [190; 680]	249 [161; 463]	0.140	
PT (sec)	13.1 [12; 14.1]	12.8 [11.9; 13.5]	0.786	
CK (U/L)	210 [48; 456]	62 [34; 127]	**0.001**	1.003 [1.001; 1.006]
CK-MB (U/L)	13 [9; 28]	10 [5; 14]	**0.05**	1.036 [0.997; 1.076]
LDH (U/L)	460 [346; 670]	354.5 [284.2; 498.7]	**0.009**	1.003 [1.001; 1.006]
ALT (U/L)	42 [26; 87]	52.5 [30; 101.7]	0.292	
AST (U/L)	52 [38; 82]	50 [34; 77.5]	0.347	
Serum creatinine (mg/dL)	1.1 [0.7; 1.2]	0.9 [0.7; 1.1]	0.197	
Serum albumin (g/L)	3.6 [3.2; 4]	3.5 [3.3; 49]	0.414	

**Table 3 medicina-58-01093-t003:** Correlations of clinical and laboratory parameters with pericardial effusion.

Clinical, Laboratory, and Imaging Characteristics	Spearman’s Rho	*p*-Value
CRP	0.201	0.05
Platelets	0.198	0.05
Myoglobin	0.408	<0.001
CK	0.325	0.001
CK MB	0.242	0.017
LDH	0.261	0.009

**Table 4 medicina-58-01093-t004:** Ventilation and admission characteristics in SARS-CoV-2 patients with and without pericardial effusion.

Ventilation and Admission Characteristics	Group A (*n*, %)	Group B (*n*, %)	*p*-Value
Nasal cannula or Venturi mask	5 (18.5%)	22 (30.1%)	0.20
Non-rebreathing masks	20 (74.1)	49 (67.1%)	
Non-invasive ventilation	1 (3.7%)	0	
Mechanically ventilation	1 (3.7%)	2 (2.8%)	
ICU admission	4 (14.8%)	3 (4.1%)	0.06
Non-ICU admission	23 (85.2%)	70 (95.9%)	

## Data Availability

The data presented in this study are available on request from the corresponding author.

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
