# Peer review of "Pericardial Involvement in Severe COVID-19 Patients"

_medicina, 2022, doi:10.3390/medicina58081093_

Round 1
Reviewer 1 Report
These are my comments regarding the manuscript entitled “Pericardial involvement in severe COVID-19 patients.” By M. Lazar et al. This is an observational cohort study. The objectives were to define the prevalence of pericardial effusions in patients with severe COVID-19 infection and its impact on mortality. Patients were divided in 2 groups: 27 with pericardial effusion and 73 without effusion. The presence of effusion was determined by CT scan of the chest. Echocardiogram was not available at the time of the study. Of the 27 patients with effusion, just 30% had clinical evidence of pericarditis. Overall, mortality was 60% higher on patients with effusions.
The topic of the study is interesting. The authors provided significant clinical data. As they stated, a significant limitation of the study is the lack of echocardiogram data. In my experience, CT scans of the chest tend to overestimate the amount of fluid present in the pericardial space. The manuscript also needs some editing. Tables provide adequate data.
Author Response
Thank you for giving us the opportunity to submit a revised draft of the manuscript “Pericardial involvement in COVID-19 patients” for publication in “Medicina”. We appreciate the time and effort that you dedicated to providing feedback on our manuscript; we have incorporated your suggestions in the manuscript.

Reviewer 2 Report
The paper is well written and contains interesting founding’s which are backed up with an extensive experimental analysis.
The only suggestion I have is to think about including graphical representation of the results.
Author Response
Thank you for giving us the opportunity to submit a revised draft of the manuscript “Pericardial involvement in COVID-19 patients” for publication in “Medicina”. We appreciate the time and effort that you dedicated to providing feedback on our manuscript; we have incorporated your suggestion into the manuscript.

Reviewer 3 Report
This is an interesting article of pericardial involvement in a cohort of COVID 19 patients. I think the topic is interesting and would be of interest to the readership of the journal. It provides information about frequency of pericardial involvement and the outcome of such patients. However, I do have the following major concerns:
1. Definition of severity- based on your selection severe patients are both intubated patient with ARDS on vasopressors as well as hypoxic patients on 4L of oxygen via nasal canula. I could argue that intubated patients are more severe and maybe they have more pericardial involvement. If possible please stratify these patients according to severity differently. Intubated vs non intubated, ICU vs non ICU admission etc
2. How did you exclude myopericarditis? patients with elevated troponin might as well have myocarditis or myopericarditis. This should be clarified. In cases of influenza myopericarditis one paper describes that patients with pericarditis had more effusions and tamponade unlike patients with myopericarditis who had higher incidence of septic shock ( please see the following paper and compare influenza vs COVID pericarditis presentations): (https://www.ncbi.nlm.nih.gov/pmc/articles/PMC9316162/)
3. Discussion needs to be improved- please add paragraph regarding complications in patients with bacterial pericarditis and compare these with your findings of viral pericarditis due to COVID 19. This will help clinician to understand relative severity of this findings compared to other common bacterial pathogens. Please also compare clinical presentations between bacterial and viral pericarditis
4. references MUST be improved. Some of them are from 1993 and 1995 and are completely irrelevant for this topic. Similar recent paper on this topic was not cited at all. Overall I think this is the major issue with the paper as I am not confident that authors have reviewed appropriate literature extensively; Please read the following new publications that are pertinent for your paper:
- https://www.ncbi.nlm.nih.gov/pmc/articles/PMC6327550/
- https://www.ncbi.nlm.nih.gov/pmc/articles/PMC8789202/
- https://pubmed.ncbi.nlm.nih.gov/32433342/
- https://pubmed.ncbi.nlm.nih.gov/35448079/
- https://journals.lww.com/jcardiovascularmedicine/Fulltext/2021/09000/Pericarditis_in_patients_with_COVID_19__a.4.aspx
Author Response
Thank you for giving us the opportunity to submit a revised draft of the manuscript “Pericardial involvement in COVID-19 patients” for publication in “Medicina”. We appreciate the time and effort that you dedicated to providing feedback on our manuscript. The authors have carefully considered the comments, and tried our best to address every one of them; we have incorporated all the suggestions into the manuscript. Please find the attached file with our point-by-point response to your comments and concerns.

Round 2
Reviewer 3 Report
I would like to thank the authors for detailed revision of the paper. In my opinion, these changes have made paper much more informative and appealing to the readers. I have no further comments.